# Solid-State Fermentation for the Recovery of Phenolic Compounds from Agro-Wastes

Nadia D. Cerda-Cejudo [1], José J. Buenrostro-Figueroa [2], Leonardo Sepúlveda-Torre [1], Cristian Torres-León [3], Mónica L. Chávez-González [1], Juan A. Ascacio-Valdés [1,*] and Cristóbal N. Aguilar [1]

[1] Bioprocesses & Bioproducts Research Group, Food Research Department, School of Chemistry, Autonomous University of Coahuila, Ing. J. Cárdenas Valdéz S/N, República, Saltillo 25280, Coahuila, Mexico
[2] Research Center for Food and Development A.C., Delicias 33088, Chihuahua, Mexico
[3] Research Center and Ethno-Biological Garden, Autonomous University of Coahuila, Unit Torreón, Viesca 27480, Coahuila, Mexico
[*] Correspondence: alberto_ascaciovaldes@uadec.edu.mx; Tel.: +52-84-4416-1238; Fax: +52-84-4416-9213

**Abstract:** Polyphenolic compounds are a group of secondary metabolites in plants; these molecules are widely distributed in fruits, vegetables, and herbs and can be found in the vacuoles of plant cells. The current trend in these compounds is their extraction to study their applications in several areas, such as the food, cosmetic, and pharmacology industry. This review article presents a critical analysis of polyphenol extraction using solid-state fermentation. The parameters of extraction, such as the substrate, temperature, pH, inoculum of the microorganism, moisture, and water activity, are discussed in detail. This biotechnological extraction method affects the concentration and recovery of polyphenolic compounds. Some polyphenolic sources that are rising for their biological properties belong to agro-industrial wastes, such as peels, seeds, and the pulp of some fruits. Solid-state fermentation is an innovative and environmentally friendly tool that can contribute to generating value-added agrifood from agro-industrial wastes.

**Keywords:** bioprocess; solid-state fermentation; phenolic compounds

## 1. Introduction

Polyphenols are biological compounds found in plants that present several health benefits. Some of these biological properties may include antiviral activity against SARS-CoV-2, anticarcinogenic, antiproliferative activity, and antimicrobial and antioxidant activities. These biocompounds can be found in fruits and vegetables, spices, teas, wines, and even dark chocolate. These polyphenols are known to possess antioxidant activities due to their capacity to neutralize harmful free radicals, prevent heart diseases and cancer, reduce inflammation, and also they may be effective against chronic diseases such as diabetes [1].

There are several types of polyphenols, such as tannins, flavonoids, anthocyanins, and polyphenolic amides, that can be found in plants, vegetables, and fruits such as pomegranates, grapes, mangoes, or apples (Figure 1). Polyphenols are more hydrophilic molecules than they are lipophilic molecules. In this way, solvents such as ethanol, methanol, acetonitrile or a mixture of these solvents with water are generally used to obtain said compounds. Concentrations of these compounds depend on various factors of the fruit, the harvesting, and the extraction method [2–4].

Traditional techniques have been used to extract phenolic compounds where the solvents used harm the environment, and higher temperatures induce negative effects on the biological properties of the extracts. Emergent technologies, or as they call them, "green technologies", have come out as an alternative to prevent damage that traditional techniques can cause; however, these emergent technologies generate an incomplete liberation of phenolic compounds. In this way, biotechnological extraction methods have come out

since the first uses of a microorganism to help degrade the cell wall with the help of enzymes causing hydrolysis, where biotransformation and biodegradation of the compounds occur [5]. Solid-state fermentation (SSF) can be implemented to obtain compounds from agro-industrial residues previously considered pollutants, such as peels or seeds, requiring small equipment and producing less wastewater [5,6]. Based on the above, this review aims to explore the advantages of extracting phenolic compounds using SSF. The factors that affect SSF are also discussed.

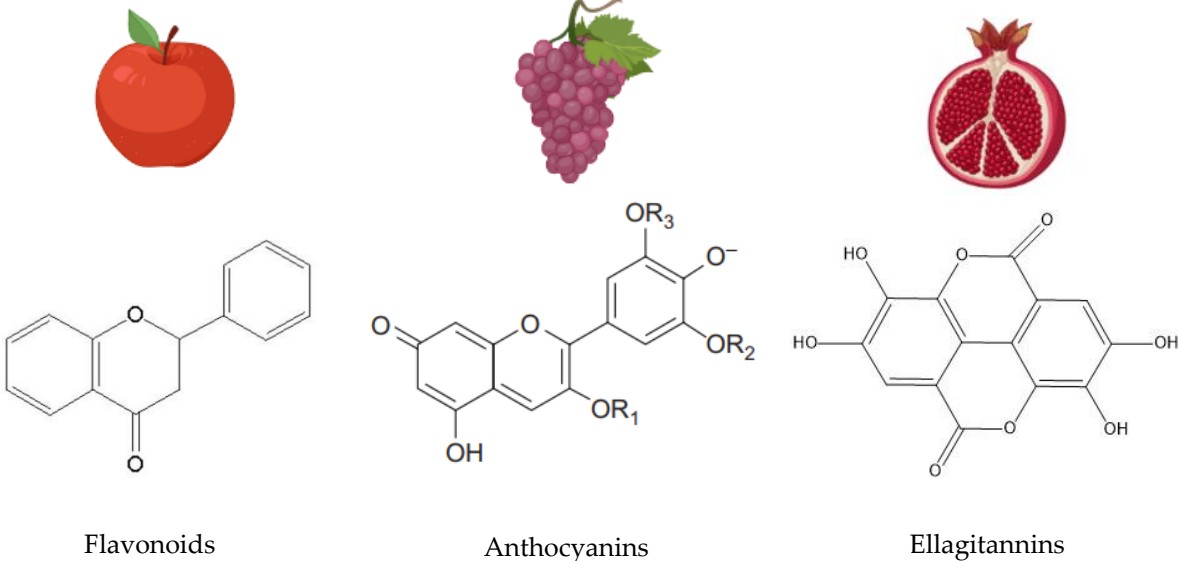

| Flavonoids | Anthocyanins | Ellagitannins |

**Figure 1.** Structure and sources of ellagitannins, flavonoids, and anthocyanins.

## 2. Polyphenol Chemistry

*Hydrolysable and Condensed Polyphenols*

Tannins are polyphenols that are divided into condensed and hydrolyzable polyphenols. Hydrolyzable polyphenols, accordant to their chemical structures, are divided into gallotannins and ellagitannins [7].

Condensed tannins are widely present in vegetables, fruits and teas—they are polyphenols conformed from two or more so-called flavan-3-ol catechetical molecules or so-called leucoanthocyan flavan-3,4-diols. They can also be the result of the union of these two types of molecules. These kinds of polyphenols do not hydrolyze under the action of diluted mineral acids. However, when boiled in water, they form insoluble compounds called plobaphenes [8].

According to Amarowicz et al. [9], gallotannins are natural polymers formed by the subsequent esterification of gallic acid and hydroxyl groups of glucose in polymeric chains, where depside bonds link the galloyl moieties. Karas et al. (2017) [10] reported that this kind of hydrolyzable polyphenols possesses antioxidant, anti-cancer, anti-ulcerative, and anti-inflammatory biological activities.

Ellagitannins are hexahydroxydiphenic acid (HHDP) esters, generally coupled with glucose with a molecular weight of 300 to 20,000 Da [11]. These molecules are mostly soluble in water and accumulate in the vacuoles of the cells, acting as protection for the plant in stressful situations and helping the plant in its adaptation to environmental changes [12]. Ellagitannins possess several biological activities, such as antioxidant, anti-microbial, or anti-cancer, that position them as compounds of interest for the health of human beings against diseases such as diabetes, cancer, and even SARS-CoV-2.

### 3. Biological Activities of Phenolic Compounds

*3.1. Polyphenols Activity against SARS-CoV-2*

Coronaviruses (CoVs) belong to the sub-family of *Ortho*-coronavirinae, under the Coronaviridae family and in order of Nidovirales, this sub-family contains alpha-, beta, gamma-, and delta-CoVs. SARS-CoV-2 is an RNA virus, and the genome sequence of this type of virus shows only moderate homology to other known coronaviruses. The viral genome encodes a protease, which plays a crucial role in the production of viral proteins and in controlling the activity of the replicase complex. Protease enzyme is necessary for virus infections and replication, making it a perfect target for designing antiviral therapies. To help stop the spread of viruses that cause diseases such as severe acute respiratory syndrome (SARS), Middle East respiratory syndrome (MERS), and human immunodeficiency virus-acquired immunodeficiency syndrome (HIV-AIDS), protease inhibitors have been developed [13,14]. For coronaviruses, the proteolytic processing of the replicase polyproteins by viral proteases must lead to the release of structural and non-structural proteins [15,16].

SARS-CoV-2 affects all people, especially those with weak immune systems and/or weak immune-based responses [13].

Polyphenols have been isolated from several kinds of plants, such as vegetables, fruits, nuts, herbs, coffee, and teas, and agro-industrial wastes, such as pomace, seeds, or peels. In the past few years, these biocompounds have attracted the attention of the scientific community since their dietary consumption has been associated with the prevention of some degenerative and chronic diseases that are significant causes of incapacity and a high mortality rate. The efficiency of their biological activities depends on their bioavailability and the number of polyphenols ingested [17–19].

According to their antiviral efficiency potential, polyphenols have been studied due to their activity against SARS-CoV-2 in cell-free polyphenol-protein interactions, cell-based virus infection, and molecular modeling studies [13,20].

In an in silico study, Singh et al. [21] found that quercetagetin, epigallocatechin gallate, and myricetin exhibited high binding affinity towards the RdRp of both SARS-CoV and SARS-CoV-2. Schetting et al. [22] reported a study where quercetin and *N*-acetylcysteine substantially nebulized the formula and relieved the respiratory symptoms of SARS-CoV-2 in a patient with antibiotics and hydroxychloroquine. In an in silico assay, Torres-León et al. [23] demonstrated that Luteolin 7-O-glucoside, Kaempferol, and Quercetin originally from Chihuahua desert plants can act as potential inhibitors against Mpro and RdRp proteins of COVID-19.

Khalifa et al. [24] describe 19 structurally different hydrolyzable tannins, such as gerannin, punicalin, and tellimagradin, among others, to find out a potent inhibitor against COVID-19 that could target the main protease of SARS-CoV-2 using in silico approaches (drug-likeness and molecular docking scan). Authors reported that hydrolyzable tannins, specifically pedunculagin, tercatain, and castalin, are efficient and selective anti-COVID-19 therapeutic compounds.

*3.2. Antioxidant Activity of Phenolic Compounds*

Antioxidant activity can be defined as the inhibition or limitation of the oxidation that occurs in proteins and lipids by restraining oxidative chain reactions [25].

The anti-oxidative characteristics of phenolic compounds have been widely studied, mostly in in vitro assays, including the inhibition of lipid oxidation, scavenging free radicals, and reduction of hydroperoxide formation.

The most common methods employed to demonstrate this particular biological activity of phenolic compounds are ferric-reducing antioxidant power (FRAP), ferrous iron chelation activity, thiobarbituric acid-reactive substances (ABTS or TBARS), 2,2-diphenil-1-picrylhydrazyl (DPPH) radical scavenging activity and the β-carotene bleaching (BCB) [26–29].

### 3.3. Antimicrobial Activity of Phenolic Compounds

Antimicrobial activity can be defined as all active agents that can inhibit the growth of microorganisms and can also prevent the formation of microbial colonies [30].

Fei et al. [31] analyzed the anti-microbial effects of olive oil polyphenols extracted against *Cronobacter sakazakii*. Authors reported a reduction of intracellular ATP concentrations, cell membrane depolarization, and a decrease in bacteria protein in *C. sakazakii* against olive oil polyphenols extract, which indicates an effective anti-microbial activity.

### 3.4. Antiproliferative and Anticarcinogenic Activities of Phenolic Compounds

Antiproliferative activity can be defined as the ability of a compound to stop the growth of the tumor cells, not allowing the cells to multiply at a fast pace.

Cancer is one of the most common causes of death around the world; some of the most common types are breast, lung and colon cancer, melanoma, and brain tumors [32].

Sung et al. [33] reported an update on the global cancer burden, estimating cancer incidence and mortality in 19.3 million cancer deaths, where female breast cancer (11.7%) has surpassed lung cancer (11.4%) as the most commonly diagnosed cancer, followed by colorectal (9.4%), liver (8.3%), and stomach (7.7%), worldwide.

In México, the most common type of cancer in females of all ages is breast cancer, affecting 28.2% of the population, followed by cervix uterine (8.9%), thyroid (8.6%), colorectum (6.6%), and corpus uterine (5.2%), among other types of cancer (42.4% between pancreas, bladder, esophagus, non-melanoma skin, and lip-oral cavity). In males of all ages, the most common incidence of cancer is prostate, present in 29.9% of the population, followed by colorectum (8.9%), stomach (5.2%), lung (5.0%), and non-Hodgkin lymphoma (4.6%), among others (46.5%) [34].

The medical industry has been using chemotherapy and radiation-based therapy as a treatment for cancer. However, the problem with these kinds of therapy is that they are really invasive, and they also tend to fill the patient with severe secondary effects that diminish the patient's quality of life. Hence, the search for new, less invasive treatments is a priority.

Proliferation is a significant part of cancer development because this can manifest in the alteration of the cell cycle related to proteins [35,36]. Cancer cells require a constant supply of oxygen and nutrients to divide themselves, like any other normal cell. Here is where polyphenol extracts take effect, causing apoptosis in cancer stem cells due to the bioactive properties of their phytochemical components [37].

## 4. Solid-State Fermentation Extraction Effects on Phenolic Contents

Solid-state fermentation is a common technique used for the production of microbial metabolites, which consists of a three-phase heterogeneous process (solid, gaseous, and liquid)—offering potential benefits for microbial cultivation for bioprocesses and product development [38]. SSF is performed on a solid substrate, mostly in agro-industrial wastes that contain a low moisture level, which is absorbed by the substrate in a solid matrix offering the transfer of oxygen that influences microbial growth [38,39].

Some of the advantages that SSF offer is a low sterility requirement, less water demand, high volume production, and the growth of aerobic and anaerobic microorganisms [40]. In addition, this bioprocess provides an alternative at a low cost for the production of several antibiotics, enzymes, biopesticides, and biosurfactants, among other bioactive compounds [41].

One of the challenges for SSF is to bring it to an industrial scale. Scaling up this bioprocess involves a variety of fundamental parameters, like moisture and water activity, temperature, inoculum, microorganism, heat, mass transfer and pH, among others, that must be carefully checked.

Mekoue et al. [42] evaluated the interactions between *Saccharomyces cerevisiae* and grape to purify polyphenols during alcoholic fermentation, and they found that polyphenols induce significant changes in the fermentation kinetics and metabolism of the microorganism

where yeast showed a lower capacity to convert sugar in ethanol, and cell mortality appear in early stages of fermentation.

Some of the effects of SSF in the extraction of polyphenolic compounds are the increase in the content of bioactive compounds found in food products, the production and extraction of these compounds from agro-industrial wastes, and the production and extraction of enzymes. Although polyphenols are present in all kinds of plants, the type and amount of these bioactive compounds vary depending on the plant, the environmental conditions, and their genetic factors. The quality of the food can be directly related to the presence of bioactive compounds that produce positive effects on human health and not only on nutritional values [43,44].

Buenrostro-Figueroa et al. [45] produced enzymatic extracts using pomegranate ellagitannins as a carbon source and inducers of ellagitannase enzyme in a SSF with *Aspergillus niger* GH1 strain. The authors reported that the content of the ellagitannase enzyme increased its values associated with ellagic acid biosynthesis. In another study, Yepes-Betancour et al. [46] reported the influence of SSF with *A. niger* GH1 strain on the release of phenolic compounds from avocado seed, which is considered an agro-industrial waste. The results revealed the ability of the fungal to degrade the compounds present in the seed, improving the antioxidant capacity and providing a broad overview of adding value to the avocado industry and developing new products.

### 4.1. Factors That Affect a SSF

#### 4.1.1. Bioreactors

A bioreactor is a container that is implemented to carry out the fermentation bioprocess, which must provide ventilation, regulation of temperature, pressure, liquid level, nourishment to the microorganism, sterilization, maintenance of sterility, and agitation if necessary. The selection of the bioreactor implemented in a SSF is of vital importance as the production of metabolites and biomass depends on this [47,48].

The material of which a bioreactor is made should possess some important properties, like tolerating the sterilization process; the material of the bioreactor should not be corrosive and should not add toxic substances to the fermentation process. A qualified bioreactor will improve optimum conditions for the microorganism to grow, like temperature and maintenance of the moisture, and provide higher productivity of metabolites in the fermentation [48,49].

In SSF, the bioreactor used is a challenge due to the low levels of moisture that it works with, and if the process involves filamentous fungi, agitation and high level of moisture, those variables could affect growth, production formation and damage the fungal hyphae. The temperature in the bioreactor must be carefully checked with the proper ventilation, with a good supply of $O_2$ and removal of $CO_2$ playing a crucial role in heat removal; the conditions mentioned before are essential for the microorganism to start and stop its metabolism [50–52].

In SSF, packed-bed bioreactors are commonly used, providing a good supply of $O_2$ to the particles when the air flows uniformly to the substrate bed, avoiding the need for mixing. At a laboratory scale, the use of packed-bed bioreactors goes from sizes up to 30–80 cm in height and 5–50 cm in diameter. Once the SSF demonstrates promising results at the pilot scale, the challenge is to scale it up to a commercial level where one of the most difficult aspects to control is the temperature [53–55]. To avoid this, a two-phase model of packed-bed bioreactors has been developed, which consists of two transfer coefficients related to the transfer of water and sensible energy across the solids-air interface [54].

#### 4.1.2. Temperature

Temperature is a factor that affects the fermentation rate because microbial production depends on it [56]. Temperature is one of the most significant factors that can influence the production of several enzymes and metabolites.

In SSF, filamentous fungus optimal growth is at 20 to 55 °C because they are mesophilic microorganisms. Nevertheless, the optimal temperature to produce metabolites of interest can be contrary to the optimal temperature for growth. Lu et al. [57] reported a study where they evaluated differences in the functional properties of microorganisms and structural characteristics between simulated natural fermentation and high-temperature fermentation in the production of soybean meal. The authors found that high-temperature fermentation inhibits the growth of microorganisms present in simulated natural fermentation and possesses a lower peptide content compared to simulated natural fermentation.

4.1.3. Inoculum and Microorganism

Inoculum can be defined as the population of cells or microorganisms that is added into the fermentation medium, and the inoculum needs to be prepared and optimized before the bioprocess can begin, as Sood et al. [58] established.

The inoculum must be optimized for a better fermentation performance, which can be conducted based on several parameters like the addition of chemicals, DNA recombination, or radiation. When it comes to microbiological techniques, inoculation of microbiological cultures is significant to obtain proper efficiency for anti-microbial sensitivity and diagnosis, both for fungal and bacterial cultures [58].

There are several criteria for inoculum preparation. These may include the morphology and physiology of the microorganisms as certain mediums have conditions that affect these parameters, such as pH, viscosity, chelating agents, or the presence of solids in the medium [59]. The microbial inoculum must be active and healthy; this means it must adapt quickly to the environmental conditions in the culture medium, which is designed for high-speed microbial growth.

The possibility of contamination in inoculum development is always present, where the repercussion is lower productivity by low microorganism development used in the inoculum preparation [60].

Microorganisms implemented in a SSF must be carefully reviewed due to their metabolism and the requirements that it demands. In SSF, several microorganisms have been used, such as filamentous fungi, bacteria, and yeast.

Yeast

Yeasts are eukaryotic organisms—generally single-celled, and reproduce by the asexual method of budding, which is adapted for specialized environments, normally liquid, that do not produce toxic secondary metabolites [61]. Yeast grows faster under severe anaerobic conditions, being capable of incrementing their number in liquid environments but limiting their development into solid surfaces. These kinds of microorganisms are used as starter cultures in bread and cheese, as well as in alcoholic fermentation products like beer or wine. Nevertheless, they also spoil some foods, such as fruits, juices, and yogurts [62]. The most common yeast species used in SSF are *Saccharomyces* spp. and *Zygosaccharomyces* spp., which are capable of growth in the complete absence of oxygen. They can continue fermenting at reduced water activities in the presence of high preservative levels and can continue fermentation under several atmospheres of the pressure of carbon dioxide ($CO_2$) [63,64].

Filamentous Fungi

Filamentous fungi are microorganisms capable of adapting to different environments and are usually employed in industrial development, used in industry as a source of alkaloids, steroids, pigments, or alcohols because fungi act as a source of several compounds in pharmaceutical-related processes [65]. They are also employed for the production of high-added value enzymes, such as cellulose, lipase, xylanase, and lactase [65]. In SSF, the best microorganism for this bioprocess is filamentous fungi, as they have the best capacity to produce industrially important enzymes. The metabolism of this microorganism means that it is possible that it can permeate the surface of the substrate in order to

transform the compounds. The most common species employed in a SSF are *Aspergillus* and *Trichoderma* (38).

De León-Medina et al. [66] evaluated Castilla rose as a potential source of polyphenols using a SSF with *Aspergillus niger*. A maximum accumulation of polyphenolic compounds was obtained at 24 h, and this study allowed the characterization of 25 different polyphenolic compounds.

Meini et al. [67] compared *Aspergillus niger* and *Aspergillus oryzae* by implementing a SSF on grape pomace as support and added substrate with tannic acid, with aimed to evaluate the enzyme activity in the recovery of polyphenols. The authors reported the production of cellulose, pectinase, and tannase enzymes. Likewise, increasing the initial moisture content increases the extraction of the total phenolic content; however, the effect of fungi fermentation on polyphenols extraction is unfavorable at low moisture levels.

Bacteria

Bacteria are the most abundant living organism on earth as there are a great variety of species and sub-species. Under optimal conditions of nutrient availability, temperature and pH, their replication is fast. Some bacteria have a symbiotic effect on the human body and grant their host many health benefits, such as intestinal microbiota. However, some are pathogenic and can be transmitted in different ways, like the ingestion of contaminated water or food, through direct contact with an infected host, or by the action of an intermediate host [68]. In SSF, the most common bacteria used in this process are the Gram (+) species members of the family *Bacillus*, which grow under aerobic conditions, secretes catalase and forms oblong endospores. Meanwhile, *Clostridium,* an anaerobic bacteria, does not secrete catalase but forms bottle-shaped endospores and produces bio-butanol [69].

For a SSF, the level of water must be low, and even if bacteria can be implemented in this bioprocess, the level of water that they need to carry out their metabolism is high compared to other microorganisms like filamentous fungi.

Ong et al. [70] used bacteria Gram (+) *Bacillus subtilis* and filamentous fungus *Aspergillus oryzae* in a SSF of okara and brewer's spent grains, where they compared these microorganisms in a mixed culture harnessing the synergistic effect through cooperative metabolism against pure cultures, with the aim of determining the total phenolic content. They reported a high increase in total phenolic content, providing that a mixed culture was effective and the cooperative metabolisms of both microorganisms were present.

4.1.4. Moisture and Water Activity

Water plays different roles in a matrix, acting as a reactant or as a solvent, facilitating the reaction in this way. Moisture content and water activity are parameters used to control SSF [71]. Water absorption capacity (WAC) determines the interaction of the substrate's macromolecules with the water allowing the formation of the gel, which is identified as how the water molecules bound with the substrate by the accessibility of their hydrophilic groups. To control, monitor, and optimize a SSF, one of the main prerequisites is to understand the water dynamics. It can be divided into three categories: overall water content, change of the whole substrate matrix by evaporation and respiration, a different state such as free or bound water, and internal water distribution transfer over the substrate matrix due to gradients [72,73].

These three dynamics affect the physical properties of the substrate, enzymatic activities, microbial physiology, and the SSF performance in general [72].

In SSF, the substrates that possess a high WAC are the most qualified to be implemented in this bioprocess because it helps the microorganism grow and develop [66]. Water activity is a key parameter for regulating microbial growth [74]. The inadequate water content will affect its biological state or distinct microbial growth, devaluating the process and destroying the microbial culture. Fungi and yeast are the microorganisms that are more appropriate for a SSF performance [75].

### 4.1.5. pH

Glab et al. [76] define pH as the parameter that is most often measured in analytical chemistry due to its effects on the position of the chemical equilibrium of the chemical reactions in aqueous solutions and other mediums. This parameter also impacts reaction kinetics, acting as a catalyst and changing the rate.

The most important regulating parameter in glucose fermentation is Ph [77]. Temudo et al. [78] reported the product spectrum of glucose fermentation as a function of the Ph in mixed culture fermentation. The authors showed that a reactor fed with glucose running under substrate limitation would shift its product distribution from butyrate, acetate, and molecular hydrogen at low pH (4–6.5) into acetate, ethanol, and formate at high pH (6.5–8.5) [79].

### 4.1.6. Substrate

A solid substrate is used as a matrix in SSF, which can be natural or synthetic. When it is a natural matrix, agro-industrial wastes are the most common organic material used as a carbon source for a microorganism. When it is an inert material, it is impregnated with a substance rich in nutrients for the microorganism to grow [80].

In SSF, homogenous and porous substrates induce enzyme production, while the heterogeneous nature of the substrate generates issues during enzyme production. The microorganism employed in SSF must also be considered in order of the substrate because when a filamentous fungus is used, the mycelium penetrates the substrate to convert it; meanwhile, yeast and bacteria grow on the surface of the substrate [38].

Buenrostro-Figueroa et al. reported the use of several inert supports, such as polyurethane foam, perlite, and nylon fiber, in the production of ellagic acid with partially purified polyphenols of pomegranate in SSF. Authors reported that the best support was polyurethane foam with the highest ellagic acid production at 24 h (231.22 mg $g^{-1}$), which promotes a better transport of nutrients and the best solubility, improving the availability of nutrients for the microorganism due to its water absorption index.

Table 1 shows several types of natural agro-industrial waste substrates used in SSF, the fermentation conditions, and the polyphenolic compounds recovered for each one.

**Table 1.** Types of natural agro-industrial waste substrates used in FES.

| Substrate | Microorganism | Reactors | Conditions of Fermentation | Polyphenolic Compounds Recovered | References |
|---|---|---|---|---|---|
| Pomegranate peel | *Aspergillus niger* PSH | Tray reactor (40 × 30 × 6 cm) | $2 \times 10^7$ spores/g at 30 °C for 18 h | Pullicalagin, punicalin, ellagic acid | [7] |
| Mango Ataulfo seed | *Aspergillus niger* GH1 | Petri dishes | $2 \times 10^7$ spores/g at 30 °C for | Gallic acid, ellagic acid | [5] |
| Grape pomace and wheat bran | *Aspergillus niger* 3T5B8 | Erlenmeyer flasks (125 mL) | $10^7$ spores at 37 °C kinetic until 96 h | Ellagitannins, anthocyanins, proanthocyanidins | [81] |
| Rambutan peel | *Aspergillus niger* GH1 | Polypropylene flask (five cubic centimeters) | $2 \times 10^7$ spores/g at 25 °C for 24 h | Ellagic acid | [82] |
| Castilla Rose | *Aspergillus niger* GH1 | Erlenmeyer flasks (250 mL) | $2 \times 10^6$ esp/g at 25 °C for 24 h | Ellagic acid, Catechin, Epicatequin, Kaempferol 3,7-O-diglucoside, | [66] |

### 4.1.7. Generalities of Natural Agro-Industrial Wastes Substrates Used in SSF Castilla Rose

Castilla rose has been used as traditional folk medicine in Mexico for stomach diseases and as an ingredient in food preparations, which makes it a potential source of phytochemicals. It is native to the semi-desert area in the northeast of Mexico, in the states of Nuevo Leon, Coahuila, and Chihuahua [83]. Castilla rose is rich in hydrolyzable and condensed polyphenols, which classifies into four groups: gallotannins, ellagitannins, complex tannins, and condensed tannins that conferee its biological and therapeutic properties.

Ueno et al. [84] reported the use of water-soluble extracts from Castilla rose in mice subjected to chronic stress in order to evaluate the anti-stress effect of the plant, reporting that it did not affect normal behavior in mice but exerted anti-stress effects under conditions of chronic stress.

Mango

This fruit is an important tropical crop because it contains a high nutritional value; it is a natural source of vitamins, essential minerals, dietary fiber, and proteins; it provides energy and possesses a unique flavor [85]. Mexico is the main producer around the globe, followed by Thailand, India, Indonesia, and China. The main phenolic compounds present in mangoes include gallic acid, ellagic acid, catechin, β-carotene, and kaempferol [86].

Mango pulp is the only consumed part of the fruit. The seed and peel are considered agro-industrial wastes containing phenolic compounds and other nutritional value compounds like starch. Chen et al. [87] evaluated the effect on the digestibility and quality of bread by adding mango peel flour, finding that it affected the quality of bread and the digestibility of starch.

Grapes

*Vitis vinifera* L. is the most common species of cultivated grape in the market. Grapes grow in clusters of elliptical berries that contain edible or nonedible seeds and can also be seedless. They are usually consumed fresh or in processed products such as wine, juice, jelly, or grape-seed oil, among others [88]. This fruit is the most cultivated crop in the world, and approximately 75% of the total production is used in the wine industry, where 20-30% of grape ends up as pomace, which contains residual pulp, seed, stem, and small pieces of stalks from the wine fermentation process [89].

Grape pomace is rich in polyphenols. The major compounds present in the seed of grapes are gallic acid, catechin and epicatechin. The major phenolic compounds found in the grape peel are procyanidins, epicatechin, epigallocatechin, catechin, and gallic acid. These biocompounds are effective free radical scavengers and have the potential to prevent cardiovascular diseases and diabetes and possess carcinogenic activity [89].

Pomegranate

Pomegranate fruit is native to Iran, Egypt, Spain, and China and is known as the Apple of Carthage or the Chinese apple. It is a sweet fruit with an acidic juice and is known for its chemo-preventive medicinal properties. It is widely used in food products such as juices, and it is known as one of nature's most powerful antioxidants as it has three times the antioxidant activity than that of green tea or red wine [90]. Pomegranate juice strengthens the liver, heart and kidney functions, increasing the body's resistance to infections and is also used in syrups and jellies [90].

After the extraction of pomegranate juice from the fruit and the separation of the seeds, the two main agro-industrial wastes produced are the peel and the seed, which are excellent sources of important bioactive compounds such as tannins, flavonoids, sterols, vitamins, minerals, and dietary fibers. These by-products have been used in the production of industrial enzymes or single-cell proteins [91]. Punicalin, punicalagin, gallic acid, and ellagic acid; also, caffeic acid, ferulic acid catechin, quercetin, and epicatechin are the phenolic compounds present in pomegranate. These bio compounds are capable of inhibiting atherosclerosis progression and reducing macrophage oxidative stress [92]. In Figure 2, the chemical composition of pomegranate is shown, where nutrients such as carbohydrates, vitamins, fats, proteins, and minerals are essential nutrients of this fruit [93,94].

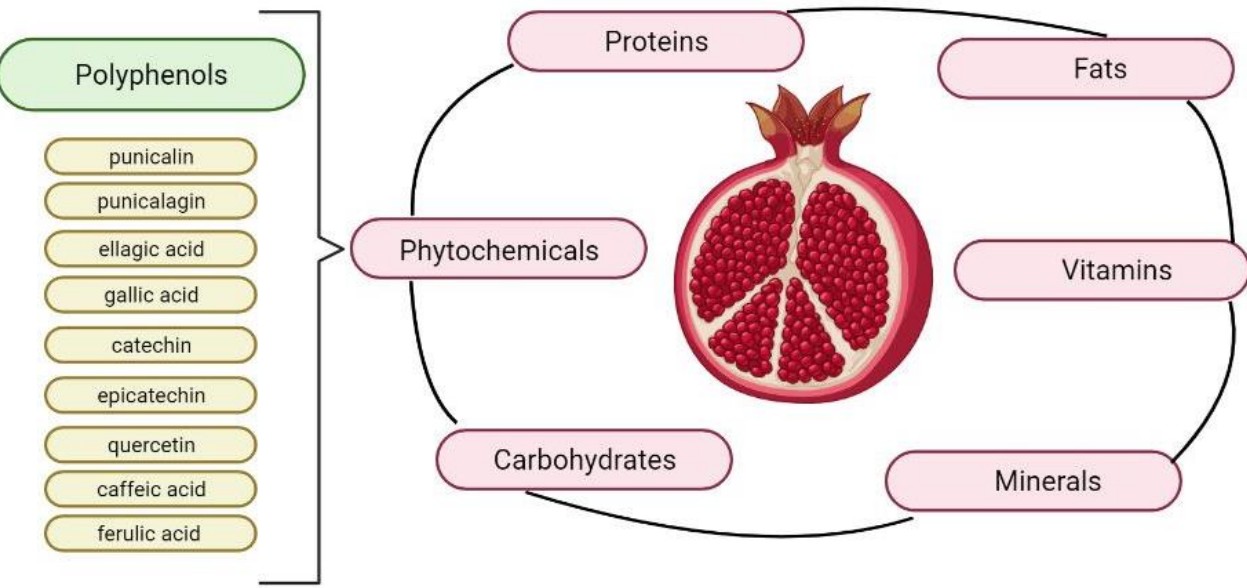

**Figure 2.** Composition of pomegranate.

Rambutan

Rambutan (*Nephelium lappaceum*) belongs to the Sapindaceae family, and it is native to Southeast Asia, and harvested especially in Malaysia, Thailand, Cambodia, and Indonesia. Rambutan is an exotic tropical fruit where the only edible part is the pulp, and its most particular characteristic is its peel—which is covered whit soft thorns that vary between colors yellow, orange, red, green and brown [95,96].

Although this fruit has been cultivated for a long time in Asia, it has a short history in Latin America, being introduced in Mexico between the 1950 and 1960, where the largest producer is the state of Chiapas, specifically in the region of Soconusco, other productor states are Tabasco and Oaxaca. In the year 2021, Chiapas reported production of 10,614 tons (96.4%), followed by Oaxaca with 194 tons (1.8%) and Tabasco with 188 t (1.7%), representing 99.9% of the national production [97].

The ethanolic extract of rambutan peel turns out to be secure to be implemented in skin tone clarifiers and antiaging cosmetic products due to the protection in the human skin fibroblasts from oxidative damage and the capacity to suppress the production of melanin in B16F10 melanoma cells inhibiting the tyrosinase enzyme and TRP-2 [98,99]. The main compounds isolated and identified from rambutan are shown in Table 2.

**Table 2.** Major phenolic compounds present in Rambutan.

| Group | Structure | Compounds | Molecular Weight (g/mol) | References |
|---|---|---|---|---|
| Ellagitannin | | Ellagic acid | 302 | [100–102] |

**Table 2.** *Cont.*

| Group | Structure | Compounds | Molecular Weight (g/mol) | References |
|---|---|---|---|---|
| Ellagitannin |  | Gerannin | 952 | [100,101,103,104] |
| Ellagitannin |  | Corilagin | 634 | [100–102,104,105] |
| Flavonoids |  | Catechin | 289 | [101,106] |
| Hydroxybenzoic acid |  | Caffeic acid | 180 | [106] |
| Hydroxybenzoic acid |  | Syringic acid | 197 | [101,106] |

## 5. Synthesis of Phenolic Compounds Recovered by SSF Assistant Extraction

In several studies across time, tannase has been reported as responsible for ellagitannin and gallotannin hydrolysis, producing ellagic acid and gallic acid, but the advanced enzymatic degradation of ellagitannin studies have reported that this enzyme is not responsible for the EA accumulation [107–110].

*Aspergillus niger* GH1 has been reported as a tannin-degrading fungal strain with a high capacity to degrade high molecular tannins into small molecules such as ellagic acid and gallic acid, using and producing tannase and ellagitannase enzymes [109,111].

Ellagitannin acyl hydrolase (EAH), known as ellagitannase, is an enzyme responsible for EA biosynthesis through ellagitannin's biodegradation [45]. Several parameters such as temperature, pH, aeration rate, nitrogen and carbon source, medium composition and packing density have been explored for its production by SSF [45,109,111,112].

Ascacio-Valdés et al. [113] reported the fungal biodegradation pathway of pomegranate ellagitannins during SSF and identified the role of the enzyme produced by *A. niger* GH1 during the ellagitannin degradation process to identify compounds produced in the recovery of EA. The authors reported that this fungal strain produces an ellagitannase enzyme induced by ellagitannins in the substrate during SSF.

## 6. Concluding Remarks and Future Perspectives

SSF as a model for the extraction of biocompounds, such as polyphenols, and how it affects their extraction performance, and the yield obtained is revolutionizing the way of doing science so that it can be applied to beneficial products for humans.

The factors of moisture, water activity, temperature, pH, inoculum, type of microorganism, as well as the substrate and type of bioreactor, are parameters that must be carefully reviewed as the success in extracting the compounds of interest depends on these variables.

SSF has the advantage of harnessing agro-industrial residues such as peels or seeds, which possess polyphenols, apart from the pulp of the fruits. With the help of enzymes such as ellagitannase present in the metabolism of microorganisms, SSF can break the chains of chemical structures and biotransform them in other compounds, recovering a greater quantity of biological compounds.

This new biotechnology is on the rise in the industry as well as in the research area since it does not negatively affect the recovered polyphenols but rather allows for better extraction yields.

**Author Contributions:** Conceptualization, J.A.A.-V.; methodology, N.D.C.-C., J.J.B.-F., L.S.-T. and C.T.-L.; formal analysis, M.L.C.-G., C.N.A. and J.A.A.-V.; investigation, J.A.A.-V. and C.N.A.; writing—original draft preparation, N.D.C.-C.; writing—review and editing, N.D.C.-C., J.A.A.-V. and J.J.B.-F.; supervision, J.A.A.-V., L.S.-T. All authors have read and agreed to the published version of the manuscript.

**Funding:** This research was funded by the Autonomous University of Coahuila, Mexico. Nadia D. Cerda-Cejudo received a scholarship from CONACyT for his postgraduate studies.

**Institutional Review Board Statement:** Not applicable.

**Informed Consent Statement:** Not applicable.

**Data Availability Statement:** Not applicable.

**Conflicts of Interest:** The authors declare no conflict of interest.

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
