# Peer review of "Solid-State Fermentation for the Recovery of Phenolic Compounds from Agro-Wastes"

_resources, doi:10.3390/resources12030036_

Round 1

Reviewer 1 Report

Resources

Recovery of phenolic compounds using SSF

The manuscript provided extensive information regarding polyphenol compounds, unlike the contents that can associate easily with the title. When looking at the title, the manuscript seemed to be about how to recover phenolic compounds using SSF. However, a considerable description of the polyphenol’s functionality occupied the whole manuscript; moreover, the fruit/plant’s general characteristics were described without mentioning recovery techniques for the SSF cases. The logical flow of the manuscript needs to be apparent. Therefore, the manuscript should show a tight linkage between ‘the recovery of phenolic compounds and SSF Techniques applied to plants/Fruits. As well, this manuscript needs to be improved by mentioning various detailed cases of applying SSF to fruits or plants. In addition to this, many typographical errors were found, and casual sentences were often observed. Therefore, the manuscript should be improved by correcting errors and using a typical journal format. 

Critical review points

1. Page 1 line 2: There is a typo in title; ‘compunds’ must be fixed to ‘compounds’
2. Page 2 line 84: After ‘at the boil,’ It may be needed to add subject (for example: it or that)

3. Page 3 line 93: In “Water’ soluble” part, It is an error in grammar. People who read can understand what it is saying, but in formal, that would be considered an error. “water’s soluble”, “soluble in water” or just “water soluble” is better expression in formal than that in manuscript.

4. Page 3 line 104: Proteasa is an typo of protease

5. Page 3 line 110: in “those who had weak immune” sentence, It may be better expression in present form not past form. Because it is more relevant with people who have weak immune now, not had in the past.

6. Page 3 line 124, the sentence connection is not smooth. If a subject is used in a preposition sentence, It should be expressed as a pronoun outside the sentence. In manuscript, it is so colloquial form.

7. Page 3 line 135: It is an error with parenthesis mark.

8. Page 4 line 152, 155: “extract” must be expressed in passive form. phenols are not extract. Those are extracted.

9. Page 5 line 214: If “considered” is used as verb same as “reported”, It must be connected with ‘and’, not apostrophe. Or if it describe about ‘avocado seed’, It is needed to delete.

10. Page 5 line 223: In the first of line, ‘of’ is not necessary in this sentence.

11. Page 5 line 226: It would be more suitable if ‘tolerate’ is in gerund form (tolerating)

12. Page 5 line 240: “SSFdemonstrates” is an typo and need blank. “SSF demonstrates” would be better.

13. Page 6 line 247: In “Temparature is one the most~”, it is not literary style. It is needed to add ‘of’ between ‘one’ and ‘the’. (Like, “Temparature is one of the~”)

14. Page 7 line 298: after “In SSF”, comma is needed

15. Page 9 line 387: There is no dot between ‘properties’ and ‘It’. If the sentence is over, the dot is needed. Or if it doesn’t mean like this, ‘It’ is not suitable in grammar. It should be modified to small letter.

16. Page 9 line 398, 399: If the plural form is expressed, the verb is without ‘s’ that express present singular form.

17. Page 11 line 443: If ‘it’ in “characteristic is it peel” is needed to express in possessive, ‘its’ is right expression in grammar

18. Page 12 line 460: ‘and’ between ‘moisture’ and ‘water activity’ is not needed in here. It is better if ‘and’ is when connected at the last. (e.g., 1,2,3 and 4. Not 1,2 and 3,4)

Author Response

Reviewer 1

The manuscript provided extensive information regarding polyphenol compounds, unlike the contents that can associate easily with the title. When looking at the title, the manuscript seemed to be about how to recover phenolic compounds using SSF. However, a considerable description of the polyphenol’s functionality occupied the whole manuscript; moreover, the fruit/plant’s general characteristics were described without mentioning recovery techniques for the SSF cases.

R= in order to attend this comment, we decided to change the tittle of the manuscript so it can be associated with the content described.

The logical flow of the manuscript needs to be apparent. Therefore, the manuscript should show a tight linkage between ‘the recovery of phenolic compounds and SSF Techniques applied to plants/Fruits. As well, this manuscript needs to be improved by mentioning various detailed cases of applying SSF to fruits or plants.

R= Attended

In addition to this, many typographical errors were found, and casual sentences were often observed. Therefore, the manuscript should be improved by correcting errors and using a typical journal format. 

R= grammar was supervised and corrected

Critical review points

  1. Page 1 line 2: There is a typo in title; ‘compunds’ must be fixed to ‘compounds’

R= grammar was corrected

  1. Page 2 line 84: After ‘at the boil,’ It may be needed to add subject (for example: it or that)

R= grammar was corrected

  1. Page 3 line 93: In “Water’ soluble” part, It is an error in grammar. People who read can understand what it is saying, but in formal, that would be considered an error. “water’s soluble”, “soluble in water” or just “water soluble” is better expression in formal than that in manuscript.

R= grammar was corrected

  1. Page 3 line 104: Proteasa is an typo of protease

R= grammar was corrected

  1. Page 3 line 110: in “those who had weak immune” sentence, It may be better expression in present form not past form. Because it is more relevant with people who have weak immune now, not had in the past.

R= grammar was corrected

  1. Page 3 line 124, the sentence connection is not smooth. If a subject is used in a preposition sentence, It should be expressed as a pronoun outside the sentence. In manuscript, it is so colloquial form.
  2. Page 3 line 135: It is an error with parenthesis mark.

R= grammar was corrected

  1. Page 4 line 152, 155: “extract” must be expressed in passive form. phenols are not extract. Those are extracted.

R= grammar was corrected

  1. Page 5 line 214: If “considered” is used as verb same as “reported”, It must be connected with ‘and’, not apostrophe. Or if it describe about ‘avocado seed’, It is needed to delete.

R= grammar was corrected

  1. Page 5 line 223: In the first of line, ‘of’ is not necessary in this sentence.
  2. Page 5 line 226: It would be more suitable if ‘tolerate’ is in gerund form (tolerating)

R= grammar was corrected

  1. Page 5 line 240: “SSFdemonstrates” is an typo and need blank. “SSF demonstrates” would be better.

R= grammar was corrected

  1. Page 6 line 247: In “Temparature is one the most~”, it is not literary style. It is needed to add ‘of’ between ‘one’ and ‘the’. (Like, “Temparature is one of the~”)

R= grammar was corrected

  1. Page 7 line 298: after “In SSF”, comma is needed

R= grammar was corrected

  1. Page 9 line 387: There is no dot between ‘properties’ and ‘It’. If the sentence is over, the dot is needed. Or if it doesn’t mean like this, ‘It’ is not suitable in grammar. It should be modified to small letter.

R= grammar was corrected

  1. Page 9 line 398, 399: If the plural form is expressed, the verb is without ‘s’ that express present singular form.

R= grammar was corrected

  1. Page 11 line 443: If ‘it’ in “characteristic is it peel” is needed to express in possessive, ‘its’ is right expression in grammar

 R= grammar was corrected

  1. Page 12 line 460: ‘and’ between ‘moisture’ and ‘water activity’ is not needed in here. It is better if ‘and’ is when connected at the last. (e.g., 1,2,3 and 4. Not 1,2 and 3,4)

R= grammar was corrected

Reviewer 2 Report

Review report on the article

 Recovery of phenolic compunds using solid-state fermentation

The topic discussed is interesting and hence the article can be considered for publication. However, there is need for thorough revision of the article for better presentation and clarity. The language needs thorough editing for better reading and to avoid ambiguity

Some suggestions are indicated below

#32. Please change ‘Regularly consumed from’ to ‘available from’

#34. Take ‘after ‘and also prevent heart diseases and cancer’ after diabetes’

Line #38 For better flow, shift before line #36, beginning of the para. Remove ;usually’

$42. Delete ‘ implemented to obtained them’

#61 ‘Hydrolysable: please remove since you have not discussed what is hydrolysable

#70. Give full  name for SSF, since you are mentioning it first in the text. Afterwards, it can be SSF.

#77, Briefly indicate what are tannins

# 80 Change ‘consumed’ to ‘present’

#84. Change ‘at the boil’ to ‘when boiled in water’

#104. Proteasa. Spelling

      There is need to bifurcate sections on sources of polyphenols as Section 2.1 (including the lines starting #112) and their biological activities (Section 2.2 or separate section, #3.0) This will allow better flow and clarity

#124. Delete ; reported by’

#180  I would have liked a more detailed discussion on the advantages of solid state fermentation in  comparison with conventional liquid fermentation preferably with a Table

#196-197. Poor sentence construction

#218. The discussion can start as ‘Factors that influence SSF. Take Bioreactors  to Section 4.8 (before Conclusion)

#240 SSFdemonstratese Not clear

#249 grows at an interval of 20 to 55 °C Change to grows optimally at  20 to 55 °C

#457-462. Poor construction of sentence

#455 Change ‘majority’ to major

#463. Why two ‘advantages’ in one sentence? Modify

Table 1.

The second column may be shifted to the last, before References

 An additional Table (Table 3) on biological activites of various polyphenols may be considered. Otherwise the information on biological activities can be given along with Table 2

Author Response

Resources

Reviewer 2

Recovery of phenolic compunds using solid-state fermentation

The topic discussed is interesting and hence the article can be considered for publication. However, there is need for thorough revision of the article for better presentation and clarity. The language needs thorough editing for better reading and to avoid ambiguity

Some suggestions are indicated below

#32. Please change ‘Regularly consumed from’ to ‘available from’

R= grammar was corrected

#34. Take ‘after ‘and also prevent heart diseases and cancer’ after diabetes’

Line #38 For better flow, shift before line #36, beginning of the para. Remove ;usually’

R= grammar was corrected

$42. Delete ‘ implemented to obtained them’

R= grammar was corrected

#61 ‘Hydrolysable: please remove since you have not discussed what is hydrolysable

R= grammar was corrected

#70. Give full  name for SSF, since you are mentioning it first in the text. Afterwards, it can be SSF.

R= grammar was corrected

#77, Briefly indicate what are tannins

R= explanation added

# 80 Change ‘consumed’ to ‘present’

R= grammar was corrected

#84. Change ‘at the boil’ to ‘when boiled in water’

R= grammar was corrected

#104. Proteasa. Spelling

R= grammar was corrected

      There is need to bifurcate sections on sources of polyphenols as Section 2.1 (including the lines starting #112) and their biological activities (Section 2.2 or separate section, #3.0) This will allow better flow and clarity

R= sections were modified to allowed flow and clarity

#124. Delete ; reported by’

R= grammar was corrected

#180  I would have liked a more detailed discussion on the advantages of solid state fermentation in  comparison with conventional liquid fermentation preferably with a Table

R= Although liquid fermentation has many advantages in phenolic compound extraction, this work focuses only on solid-state fermentation as well as the factors that affect it.

#196-197. Poor sentence construction

R= sentence was corrected

#218. The discussion can start as ‘Factors that influence SSF. Take Bioreactors  to Section 4.8 (before Conclusion)

#240 SSF demonstratese Not clear

R= grammar was corrected

#249 grows at an interval of 20 to 55 °C Change to grows optimally at  20 to 55 °C

R= grammar was corrected

#457-462. Poor construction of sentence

#455 Change ‘majority’ to major

R= grammar was corrected

#463. Why two ‘advantages’ in one sentence? Modify

R= advantages modified

Table 1.

The second column may be shifted to the last, before References

R= table structure was changed.

 An additional Table (Table 3) on biological activites of various polyphenols may be considered. Otherwise the information on biological activities can be given along with Table 2

R= biological activities of polyphenols in general are specified in section 3.

Reviewer 3 Report

This review lacks novelty. Need to improve a lot.

When ever there is a quantitative data mentioned, it should be supported with a reference. Here it is missing.

Do make a table with the compounds, sources, biological functions and references.

Authors need to support why they are advocating for SSF. A synthesis is needed, which is poorly written.

In biorectors section, depict with figures about general bioreactor configuration and mention about what are the changes need to be made for varied microbes.

Add a brief note on microbes, their mode of fermentation, advantages, disadvantages and any commercial studies so far reported.

Critically evaluate about the de-merits, interventions to be made, and future prospects

Re-write the conclusion portion

English language is very poor and need to improve.

All references and sub heads formatting need to be formatted as per journal style.

Author Response

Resources

Reviewer 3

This review lacks novelty. Need to improve a lot.

When ever there is a quantitative data mentioned, it should be supported with a reference. Here it is missing.

R= references added

Do make a table with the compounds, sources, biological functions and references.

R= compounds mentioned are available in section 2, sources are mentioned in section 4.1.7, biological activities in general are mentioned in section 3.

Authors need to support why they are advocating for SSF. A synthesis is needed, which is poorly written.

R= synthesis was written at section 5

In biorectors section, depict with figures about general bioreactor configuration and mention about what are the changes need to be made for varied microbes.

R= bioreactors figure was added

Add a brief note on microbes, their mode of fermentation, advantages, disadvantages and any commercial studies so far reported.

R= generalities of microbes used in SSF are mentioned in section 4.1.3

Critically evaluate about the de-merits, interventions to be made, and future prospects

Re-write the conclusion portion

English language is very poor and need to improve.

R= grammar was supervised and corrected

All references and sub heads formatting need to be formatted as per journal style.

R= all references were adapted to the format of the journal R= some points were added

Round 2

Reviewer 2 Report

Title:

Recovery of phenolic compounds using solid-state  fermentation Effect of factors of solid-state fermentation in the 3 recovery of polyphenolic compounds from agro industrial wastes

The title could have been: Solid-state fermentation for the recovery of phenolic compounds from agro wastes

While the article is informative, I have serious concerns with the language. Although the authors have attempted to correct the presentation, there are several cases of poor sentence construction and presentation. These need to be rectified before publication. I suggest the article may be checked for editorial improvements by a professional editorial agency.

A few corrections are suggested below: (There are many)

Abstract

The current trend of this kind of compound is the extraction to study their applications (Poor sentence)

Lines 101-102. Can be deleted.

311-312.  ‘employed in the industrial development, used in industry’ Poor construction. What is meant by ‘adaptable microorganisms’?

318. What is meant by ‘dare say’?

484. Delete ‘Synthesis’ and also ‘using’ and ‘extraction’ in the sentence.

Author Response

Reviewer 2

Title:

Recovery of phenolic compounds using solid-state  fermentation Effect of factors of solid-state fermentation in the 3 recovery of polyphenolic compounds from agro industrial wastes

The title could have been: Solid-state fermentation for the recovery of phenolic compounds from agro wastes

R- Ttittle corrected

While the article is informative, I have serious concerns with the language. Although the authors have attempted to correct the presentation, there are several cases of poor sentence construction and presentation. These need to be rectified before publication. I suggest the article may be checked for editorial improvements by a professional editorial agency.

R- Grammar was checked by the app Grammarly

A few corrections are suggested below: (There are many)

Abstract

The current trend of this kind of compound is the extraction to study their applications (Poor sentence)

R- Grammar corrected

Lines 101-102. Can be deleted.R- Grammar corrected

311-312.  ‘employed in the industrial development, used in industry’ Poor construction. What is meant by ‘adaptable microorganisms’?

R1- Grammar corrected

R2- Adaptable microorganism means that it can modify itself to the environment in order to survive in the adversity that it presents (moisture, temperature, ph, etc.).

  1. What is meant by ‘dare say’?

R- In the sentence, refferes that is betting the microorganism used is the best for this bioprocess according to the literature.

  1. Delete ‘Synthesis’ and also ‘using’ and ‘extraction’ in the sentence.

R- Grammar corrected

Reviewer 3 Report

Suggested corrections incorporated. Paper may consider for acceptance.

Author Response

Suggested corrections incorporated.

Round 3

Reviewer 2 Report

The article is interesting. I would suggest extensive language editing before publication of the article.

Author Response

Reviewers’ comments

The article is interesting. I would suggest extensive language editing before publication of the article.

R= English language have been re-edited
